# What Factors Are Associated with Attitudes towards HPV Vaccination among Kazakhstani Women? Exploratory Analysis of Cross-Sectional Survey Data

**DOI:** 10.3390/vaccines10050824

**Published:** 2022-05-23

**Authors:** Gulzhanat Aimagambetova, Aisha Babi, Torgyn Issa, Alpamys Issanov

**Affiliations:** 1Department of Biomedical Sciences, School of Medicine, Nazarbayev University, Nur-Sultan 010000, Kazakhstan; aisha.mukushova@nu.edu.kz (A.B.); tshokanbaeva@nu.edu.kz (T.I.); 2Department of Medicine, School of Medicine, Nazarbayev University, Nur-Sultan 010000, Kazakhstan; alpamys.issanov@nu.edu.kz; 3School of Population and Public Health, The University of British Columbia, Vancouver, BC V6T1Z8, Canada

**Keywords:** HPV, HPV vaccine, HPV vaccine knowledge, HPV vaccine awareness, Kazakhstan, cervical cancer prevention

## Abstract

**Background.** The high prevalence of HPV infection among Kazakhstani women and the absence of an HPV vaccination program are directly reflected in increasing rates of cervical cancer incidence and mortality. Kazakhstan made its first attempt at introducing the HPV vaccine in 2013, but was unsuccessful due to complications and low public acceptance. The attitudes of Kazakhstani women towards the vaccine were never measured. Therefore, this study aims to investigate the attitudes of women towards the HPV vaccine and determine factors associated with positive, negative, or neutral attitudes. **Methods.** A 29-item survey consisting of 21 demographic and contextual questions and 8 Likert-scale questions was distributed among women attending gynecological offices in four major cities of Kazakhstan from December 2021 until February 2022. Attitudes of women were measured based on their answers to the eight Likert-scale questions. Ordinal logistic regression was built to find associations between demographic characteristics and attitudes of women. **Results.** Two hundred thirty-three women were included in the final analysis. A total of 54% of women had positive attitudes towards the vaccine. The majority of women did not trust or had a neutral attitude towards the government, pharmaceutical industry, and traditional and alternative media. However, the trust of women was high in medical workers and scientific researchers. Women’s age, education, number of children, effect of the 2013 HPV program, and trust in alternative medicine were included in the ordinal logistic model. Women with a low level of education, a high number of children, who believe in alternative medicine, and who were affected by the failed 2013 vaccination program were less likely to have a positive attitude towards the vaccine. **Conclusions.** Contrary attitudes towards HPV vaccination exist among Kazakhstani women, with approximately half having positive and almost half having negative or neutral attitudes towards the vaccine. An informational campaign that takes into consideration women’s levels of trust in different agencies, as well as targets those who are the most uninformed, might help in a successful relaunch of the HPV vaccination program. However, more studies that cover a higher number of women are required.

## 1. Introduction

High-risk human papillomavirus (HR-HPV) infections cause a wide variety of benign and malignant conditions, including cervical cancer [1,2,3,4]. More than 90% of cervical cancer cases are attributed to HR-HPV infections, with HPV-16 and HPV-18 being reported to cause 70–75% of cases [3,4]. The knowledge that persistent HR-HPV infection is causally associated with cervical cancer has resulted in the development of prophylactic vaccines to prevent HPV infection and, thus, decrease the rates of HPV-related diseases [4]. HPV vaccination programs have been implemented successfully in many high-income countries and have led to a decline in cervical cancer incidence rates [5,6,7]. However, the situation in low-income countries remains deplorable—cervical cancer is the leading cause of cancer-related death in women, as primary and secondary prevention interventions are insufficient or lacking [4].

The association of HPV infection with anal, penile, head, and neck cancers is also well documented [8,9,10]. Therefore, immunization with the HPV vaccine can protect from the development of other HPV-related cancers.

The prevalence of HPV infection varies among countries worldwide, with the highest prevalence of HR-HPV infection among women in the developing regions of Southern and Eastern Asia (44.4% and 36.3%, respectively) [11]. A high prevalence of HPV infection was also identified among Northern American and Eastern European populations (41.1% and 28.9%, respectively) [11]. The lowest prevalence of HPV infection is reported in Middle Eastern and North African countries (7–16%) [12]. According to the available epidemiological data, HR-HPV prevalence in Kazakhstan is high among women attending gynecological offices, ranging from 39% to 43% [13,14,15].

The high prevalence of HR-HPV strains in Kazakhstan is contributing to increasing cervical cancer rates. Over the past 10 years (2009–2018), the crude rate of cervical cancer incidence in Kazakhstan has increased from 16.3 ± 0.4 to 19.5 ± 0.5 per 100,000 female population (*p* < 0.001) [16]. The national cervical cancer screening program was updated in 2017 [17], and eligible women received the screening free of charge. However, according to reports, the state-funded coverage reached only 50% of the target population [18]; thus, the screening coverage was low (around 46%) [18,19,20].

In 2013, the HPV vaccination campaign was introduced in Kazakhstan as a pilot program in four large regions [21]. Bivalent and tetravalent HPV vaccines were approved for the campaign, targeting 11–12-year-old girls [18]. However, the lack of an HPV vaccination awareness campaign that should have preceded and accompanied the vaccination program and the influence of social media’s negative content on the program have led to a negative public reaction and unwillingness of parents to vaccinate their children against HPV [18,21,22]. Parents refused to vaccinate their daughters due to concerns about the HPV vaccine’s safety and efficacy [23]. The vaccination program was discontinued in 2015 indefinitely. In 2020, the Ministry of Healthcare of the Republic of Kazakhstan announced the intention to relaunch the HPV vaccination program; however, due to the COVID-19 pandemic, it has been postponed.

Given the importance of the reintroduction of HPV immunization to prevent new cervical cancer cases, it is essential to understand what factors are associated with parents’ willingness to vaccinate their children. Studies have shown that attitudes towards HPV vaccination are associated with HPV vaccination uptake [24]. Multiple factors play a role in shaping attitudes towards vaccination [25,26]. Education, social status, income, and cultural and religious preferences have a substantial impact on people’s beliefs, thus affecting their understanding of different aspects of HPV as a sexually transmitted infection (STI) and its related diseases [26]. As reported by studies, attitudes and intentions to receive HPV vaccination vary from 15% to 95% in different societies, depending on the multiple factors involved [26,27,28].

As one of the tasks announced by the World Health Organization (WHO) in the Global Strategy to Accelerate the Elimination of Cervical Cancer [29,30] is to increase HPV vaccination coverage up to 90% among the target group, it is very important to investigate HPV knowledge, awareness, and attitudes towards HPV vaccination. For the successful relaunching and implementation of the HPV vaccination program in Kazakhstan, the investigation of attitudes to HPV vaccination among the general population has become even more essential, especially among women, the population most affected by HPV-related cancers. Thus, by surveying Kazakhstani women across the country, this pilot study aimed to evaluate attitudes towards HPV vaccination and explore factors potentially associated with differing attitudes towards the vaccine.

## 2. Materials and Methods

### 2.1. Study Design and Subjects

This is a cross-sectional study, which was conducted from December 2021 until February 2022 in four major cities (Nur-Sultan, Almaty, Aktobe, Oskemen, Kazakhstan) representing the central, southern, western, and eastern regions of Kazakhstan. Women aged from 18 to 70 years old visiting gynecological offices were recruited for the study. A total of 399 women agreed to participate in the study, and 347 reported full demographic characteristics. Out of those 347 women, 321 (80.45%) fully answered the HPV vaccine attitude questions. Almost 27% were excluded due to incomplete answers to the contextual questions. In total, 233 women were included in the complete-case analysis (Figure 1).

### 2.2. Study Instrument

Two questionnaires were used in the study. The first collected data on demographic characteristics of women. The second questionnaire was adapted from the French Survey Questionnaire for the Determinants of HPV Vaccine Hesitancy (FSQD-HPVH) [31]. The questionnaire was adapted to the context of Kazakhstan (Appendix A). Based on experts’ opinions, some questions were removed due to being irrelevant, and equivalent questions were added. For example, statements such as “Most of my friends get their daughters vaccinated against HPV” were removed, as the HPV vaccine is not readily available in Kazakhstan. The statement “Since the controversy over the H1N1 flu vaccination, I have less confidence in the French vaccination recommendations” was replaced with a statement of COVID-19, as it fits the context. All questions were translated and back-translated to Kazakh and Russian languages (official languages of the country) by experienced independent bilingual translators. The questionnaire included 29 items asking about contextual factors (historical factors, policies, and mandates), trust in different agencies, and beliefs and attitudes towards HPV vaccination.

### 2.3. Study Variables

#### 2.3.1. Independent Variables

Independent variables were socio-economic and demographic characteristics such as age, ethnicity, level of education, city, family income, marital status, and number of children. According to age, women were categorized into four major groups: 18–27 years old, 28–33 years old, 34–43 years old, and 44 years old and older. Age of women was categorized according to the quartiles. Ethnicity was categorized into 2 groups: Kazakh and other ethnic groups. There were three levels of education: high school or below, college, and university. In January 2022, the average income in Kazakhstan was 269 149 tenge (USD ≈ 520) [32]. Income lower and higher than the average was categorized as lower and upper income, while average income was categorized as middle income.

In addition, contextual factors such as historical factors, policies, and mandates were used as independent variables. This section consisted of 6 Likert-scale questions. The major topics of the questions were the impact of COVID-19 vaccination on trust in HPV vaccine recommendations (COVID-19 effect), the 2013 HPV vaccination program in Kazakhstan (2013 HPV program), the relationship of HPV vaccination with sexual promiscuity, compulsory vaccine mandated by the government, freedom of vaccine choice for children, and alternative medicine strengthening body immune system (alternative medicine beliefs). Statements about trust in different agencies were also used as independent variables. The Likert scale was used to measure trust in agencies and health systems such as the pharmaceutical industry, government, medical workers, scientific researchers, traditional media, and alternative media.

#### 2.3.2. Outcome Variable

The attitude of the participants was measured using 3 positive statements and 5 negative statement items on the Likert scale. Negative statements were reversed, and the mean attitude score of the participants was calculated. The attitude of the women was categorized as <3—negative attitude, 3—neutral attitude, and >3—positive attitude to separate women who have a positive or negative opinion about the vaccine from those who are hesitant. It is important to differentiate those who refuse vaccination from those who do not have an opinion and lack confidence in vaccination [33].

### 2.4. Statistical Analysis

Statistical analysis was performed using STATA 16 [34]. Cronbach’s alpha was calculated for attitude items to measure internal reliability. The scale included both positive and negative items. Positive items had a Cronbach’s alpha coefficient of 0.82. Negative statements had a Cronbach’s alpha coefficient of 0.75, and the overall Cronbach’s alpha was 0.79. Factor analysis was performed to test the questionnaire’s consistency and validity. The Kaiser–Meyer–Olkin measure of sampling adequacy (0.752) and the Bartlett test of sphericity (*p* < 0.001) were performed before the factor analysis. All factors had a uniqueness lower than 0.6, except for the statement “Majority of my friends vaccinate their daughters from HPV”, which had a uniqueness of 0.81. The item was dropped from the attitude scale, as it was not relevant to the Kazakhstan context due to the HPV vaccine not being readily available in the country.

Descriptive statistics consisting of mean values, standard deviations, and frequencies were obtained using univariable and bivariable analysis. Chi-square and Fisher’s exact tests with a significance value of <0.005 were used to analyze the relationships between categorical variables.

Ordinal logistic regression was performed to explore factors associated with attitude towards HPV vaccination among women. Variables that showed significance in the bivariable analysis and were important as epidemiological factors were included in the final model. Among participants’ characteristics, variables such as age, education, and number of children were included in the final model. Although participants’ age was not a statistically significant factor, it was included in the final model due to its importance for the epidemiological picture. Two contextual factors such as distrust in the healthcare system due to complications of the HPV vaccination program in Kazakhstan in 2013 and trust in alternative medicine (alternative medicine strengthening the body’s defenses) were considered for the final model. The model assumptions were checked. The goodness of fit of the model was checked with Hosmer–Lemeshow, Pulkstenis–Robinson Chi-square, and deviance tests, as well as the Lipsitz likelihood-ratio test.

### 2.5. Ethical Considerations

The study was conducted in compliance with the Declaration of Helsinki and was approved by the Institutional Research Ethics Committee of Nazarbayev University (NU IREC) on 23 April 2019 (IREC Number: 146/4042019). Before inclusion in the study, all potential participants were informed about the aims, methods, risks, and benefits of the study. Verbal consent was received from participants after an explanation of the voluntary and anonymous nature of the study.

## 3. Results

### 3.1. Participants’ Characteristics

The social and demographic characteristics of women are shown in Table 1. The mean age of the study participants was 36.46 ± 11.18 years. More than 80% of women represented the Kazakh ethnicity. Most women (55%) had a university degree, which is in line with the official government statistics that reported the gross enrollment rate in higher education in Kazakhstan in 2020 to be 64.07%, of which 70.5% were women [32]. The majority were either married or were in a committed relationship (80%) and had one to three children (69%). There was almost an equal distribution of the place of residence among the participants, except for Aktobe city, where the number of participants was 29%. Almost half of women (48%) reported a family income lower than the national average level.

### 3.2. Contextual Factors

Table 2 demonstrates to which extent women agreed with the statements about contextual factors. Most women (48%) were neutral about COVID-19′s effect on trust in the HPV vaccine and about the 2013 HPV program’s effect on their trust in the healthcare system. More than half of women (56%) disagreed with the statement that vaccinating teenage girls against HPV encourages them to have sex. The majority of women (42%) were not in favor of the compulsory vaccines for children mandated by the government of Kazakhstan. The majority of women (44%) were neutral to the statement that everyone should be able to decide which vaccines are needed for their children. More than half of women (61%) agreed that alternative medicines strengthen the body’s defenses, thus leading to a complete cure.

### 3.3. Trust in Sources about Vaccination

The least trusted information sources among women were alternative media, the government, and the pharmaceutical industry. The most trusted were scientific researchers, closely followed by medical workers (Figure 2).

Table 3 demonstrates the extent to which women trust the sources to tell the truth about the vaccine. The most trusted sources to tell the truth about the vaccine, associated with a positive attitude towards vaccination, were healthcare professionals (60%) and scientific researchers (61%). The high level of distrust of the pharmaceutical industry was associated with a positive attitude towards HPV vaccination (58%). The government was also not viewed as a trustworthy source and was associated with a positive attitude towards HPV vaccination (61%). Both alternative (WhatsApp Messenger, Instagram, Telegram, YouTube, etc.) and traditional media (TV, newspapers) had lower trust levels among the participants with both negative and positive attitudes towards HPV vaccination. All the sources were statistically significantly associated with attitude towards HPV vaccination.

### 3.4. Attitude towards HPV Vaccination

#### 3.4.1. Bivariable Analysis

Bivariable analysis between attitude towards HPV vaccination and patient characteristics and contextual factors is shown in Table 4. A positive attitude towards HPV vaccination was reported by 52% of the respondents, with the highest proportion in the youngest age group and the lowest in the oldest age group. There was no statistical difference in ethnicities regarding attitude towards HPV vaccination. The highest proportion of positive attitudes (58%) was observed among women with a university degree and the lowest (14%) among women with only high school or unfinished high school education. Negative attitude was the lowest among women with middle income (7%) and highest among women with low income (27%), while women with an upper level of income had the highest proportion of positive attitudes (62%). The majority of women without children (72%) had a positive attitude towards the vaccine, while only 16% of women with 4 and more children had the same attitude.

The Chi-square test showed that all contextual factors were statistically significantly associated with the level of attitude towards HPV vaccination. Among women who had no prejudice that HPV vaccination encourages teenage girls to have sex, 65% had a positive attitude. Among those who disagreed with compulsory vaccination, a majority (54%) had a positive attitude to the HPV vaccine, and among those who agreed with compulsory vaccination, a majority (58%) had a positive attitude to the HPV vaccine. Similarly, a majority of women who agreed (52%) or disagreed (71%) with freedom of vaccine choice for their children were women with a positive attitude towards the HPV vaccine. The majority of women (73%) who disagree with the statement that alternative medicine strengthens the body’s defenses had a positive attitude towards the HPV vaccine.

#### 3.4.2. Ordinal Logistic Regression

Women aged between 34 and 43 were 1.23 times more likely to have a positive (in relation to neutral or negative) attitude towards HPV vaccination in comparison to women aged between 19 and 27 years, adjusting for other variables (Table 4). Having a college or university degree increased the odds of a positive attitude by 1.81-fold and 2.38-fold, respectively, compared to incomplete/complete high school education, holding other variables constant. Women with 1–3 children or 4+ children were 0.45 and 0.26 times the odds of a positive attitude towards HPV vaccination in comparison with women with no children, adjusting for other factors.

Women who were neutral about their confidence in the healthcare system were 0.34 times the odds of a positive attitude towards HPV vaccination. Agreeing that alternative medicines strengthen the body’s defenses or being neutral about the statement decreased the odds of a positive attitude by 0.41-fold and by 0.48-fold compared with women who disagreed with the statement, adjusting for other variables (Table 4).

## 4. Discussion

To date, there are no published studies investigating attitudes toward the HPV vaccine among Kazakhstani women. Thus, this is the first study that aimed to examine attitudes toward HPV vaccination and explore factors associated with a positive attitude toward HPV vaccination in Kazakhstani women. Since HR-HPV prevalence is high among women in Kazakhstan [13,14,15,21], and cervical cancer incidence has increased in the past decade [16], becoming the fourth leading cause of death from cancer among women [35], it is important to implement primary prevention of HPV infection and its related diseases in the country. However, the successful relaunching of the HPV vaccination program largely depends on HPV vaccine attitudes; therefore, studies investigating society’s attitudes towards the vaccine are an essential part of facilitating the process.

Some historical contextual factors were taken into consideration in this study, such as previous HPV vaccination program complications/failure in Kazakhstan. The situation in Japan closely resembles the failure of the HPV vaccination program in Kazakhstan. Reports of side effects in mass media and prompt cancellation of the program had a long-lasting effect in Japan, causing the vaccine coverage to fall to less than 1% [36]. In Kazakhstan, the pushback intensified after media coverage of two 11–12-year-old girls having trouble breathing, hallucinations, and being hospitalized in the ICU immediately after administration of the HPV vaccine. As was later clarified, such a reaction resulted from inappropriate actions of medical personnel, who panicked at an allergic rash to the vaccine and administered a mixture of sedatives and painkiller medications. Nevertheless, the damage was done. The media quoted government officials and medical workers saying “73 percent of participants in [HPV] vaccine trials acquired new diseases—similar facts have been recorded around the world” and “Who knows what will happen to the reproductive function of [HPV] vaccinated girls in ten years?” [37].

Among our respondents, those who were affected by the failed 2013 HPV vaccination program had a 79% lower likelihood of having a positive attitude towards the vaccine, adjusting for other variables. Overall, 16% of women indicated that the failure of the 2013 HPV vaccination program in Kazakhstan had affected their confidence in the healthcare system, and the majority of them (40%) had a negative attitude towards the HPV vaccine. Thus, an unsuccessful attempt to implement the HPV vaccination program in Kazakhstan in 2013 had a significant impact. This finding is similar to the study reporting data from Romania and France, where the initial HPV vaccination program coverage reported a poor response and required specific action plans to improve the situation [38,39].

Rapid introduction of the COVID-19 vaccine is another factor considered in this study, as it has been reported previously that the level of COVID-19 hesitancy is high in Kazakhstan [40,41], and the media coverage of the vaccine is politicized worldwide [42,43], which might carry over to other domains in healthcare, such as the HPV vaccine [44].

The ongoing debates around the COVID-19 vaccine, however, did not have the same effect. Among women who disagreed with having less confidence in HPV vaccination recommendations since the controversy over the COVID-19 vaccination, 69% still had a positive level of attitude towards HPV vaccination. Overall, 14% of women indicated that COVID-19 controversies have affected their trust in HPV vaccination, and their distribution among the attitude groups was even. This finding falls in line with previous studies that show a lack of hesitancy overlap between HPV and COVID-19 vaccines [45].

Unfortunately, this study revealed a low trust of the respondents in the government (including the Ministry of Healthcare) as the main policymaker in Kazakhstan and in pharmaceutical companies. The majority of the respondents (44%) did not trust the government; only 13% did, the rest remaining neutral. This evidence shows that the Ministry of Healthcare and other governmental agencies in Kazakhstan should reinforce the information campaign in trustworthy ways. Similarly, participants had low levels of trust in alternative media and mostly low or neutral levels of trust in traditional media. At the same time, the fact that in our study, researchers and healthcare professionals are the most trusted source of information about the vaccine gives hope and identifies directions that the Ministry of Healthcare can employ to increase HPV vaccine awareness and improve attitudes. Engaging and strengthening the relevant healthcare workforce in preparation for HPV vaccine introduction could be achieved through training/seminars, information about the evidence and necessity of HPV vaccination and cervical cancer, and communication training to address parents’ concerns and make recommendations. Our findings of trust are comparable with studies in European countries (Sweden, Hungary, France, and the UK), where mistrust of health authorities was reported by 47% to 55% of the respondents [46]. On the other hand, the results of one Italian study showed high trust in doctors (85.3%) and teachers (90.4%) [47]. Similar to our finding, the study by Karafillakis et al. (2019) also revealed mistrust of pharmaceutical companies due to their underlying profit-making motives reported in many other European countries (Bulgaria, Romania, Sweden, Ireland, the Netherlands, Spain, and the UK) [46]. The results are also somewhat comparable to those in the USA, where levels of trust were generally high for healthcare providers and low for pharmaceutical companies. The trust for governmental organizations, however, showed mixed results in varying studies. Unlike in our study, the factor of ethnicity was significant for trust levels in the USA [48]. This could be explained by a more homogenous ethnical composition of Kazakhstan.

In this study, a positive attitude towards HPV vaccination was reported by 52% of the respondents. Education, place of residence, level of income, and number of children were the factors that were found to be significantly associated with the level of attitude towards HPV vaccination. Women with high income levels had the highest proportion of positive attitudes toward the HPV vaccine (62%). On the other hand, among women with high-school-level education and those who did not finish high school, 62% were neutral towards HPV vaccination. Thus, the level of education directly correlates with positive attitudes toward the HPV vaccine. Similar results were obtained in an Italian study, where 49.7% of the respondents had a positive attitude toward the HPV vaccine [47], and in a Romanian study, where 50.7% of women have a positive attitude toward the vaccine [38]. Both cited studies, and our study, confirmed that level of awareness and attitudes were directly linked to education [38,47]. This was also confirmed in the study by Özdemir et al. (2020), where positive attitudes towards HPV and the HPV vaccine increased in employed women and those who had high education and economic levels [49].

However, having such a huge proportion of neutral respondents in our study indicates the direction of further actions. Informational interventions to increase awareness of the HPV vaccine and the advantages of being vaccinated should be developed, as a positive attitude toward the HPV vaccine is associated with a higher level of intention to receive the HPV vaccine [26,50].

Unfortunately, only 16% of mothers with four or more children had a positive attitude towards the HPV vaccine, and the majority (64%) had a neutral attitude. This is an important group of the population, who requires more information on the vaccine, as their attitude towards the vaccine has a direct impact on the HPV vaccination program coverage among the target group of 9–11 years old girls. The situation could be improved if relevant and effective informational programs are employed to explain the advantages of the HPV vaccine in HPV infection and cancer prevention, as has been demonstrated by a study among parents in Japan [51].

As was confirmed by many studies, a significant reduction in the incidence of precancerous cervical lesions and cervical cancer was observed after the introduction of HPV vaccination programs [6,52,53,54]. Moreover, HPV vaccination is associated with a substantially reduced risk of invasive cervical cancer at the population level [52]. This knowledge, together with the knowledge of HPV as the most common STI, could improve the HPV infection spread [55]. However, very few studies have reported the association between knowledge of STI and sexual behavior and HPV status [56,57]. Thus, more research on HPV infection and HPV vaccine knowledge, behavior, and attitudes is required.

Study strengths and limitations. This was the first study to assess attitudes towards HPV vaccination among Kazakhstani women. Another strength of this study is covering the female population in the big cities of four Kazakhstani regions (central, southern, eastern, and western) and investigating the factors associated with attitudes towards HPV vaccination. However, since we approached only the population of big cities, but not these regions’ rural areas, this study cannot fully represent the whole country’s female population. Moreover, in this study, the proportions of participants with university-level and college-level education were 55% and 34%, respectively. These indicators could be different if we include participants from rural areas. These are the major limitations that we hope to overcome in our future studies to obtain a more precise picture in terms of participants’ residence and education. Another limitation of this study is its cross-sectional nature and resulting lack of causality. The study employed convenience sampling with self-reporting methods. This limits the representativeness of our findings due to selection and nonresponse biases. The descriptive nature of our statistical model, as well as the low number of respondents, prompts for expansion of the study beyond the pilot investigation. Moreover, in future studies, we should investigate parents’/mothers’ attitudes towards HPV vaccination further, as they are the main decision makers if the vaccination of the target group is planned.

There are many issues related to cervical cancer prevention that require solving in Kazakhstan in the near future [18]. Although it requires attention and effort from healthcare policymakers, social media could help to improve the situation with overall societal awareness. Evidence from recent studies has indicated that social media has good potential to reach adolescents and young adults with information about HPV [58], and instead of creating a negative view, it could help to promote the HPV vaccination campaign.

## 5. Conclusions

This study shows contrary attitudes toward HPV vaccination exist among Kazakhstani women, with approximately half of women having positive and almost half having negative or neutral attitudes towards the vaccine. We found that women’s age, education, number of children, confidence in the healthcare system, and belief in alternative medicine were associated with attitudes towards HPV vaccination. These factors, as well as high levels of trust towards medical and science workers, should be taken into consideration when planning context-specific health educational interventions to form positive attitudes towards HPV vaccination in Kazakhstani women. In addition, sharing accurate information regarding the safety, effectiveness, and benefits of HPV vaccination to prevent HPV infection and related diseases could potentially improve women’s attitudes. HPV educational interventions are warranted to successfully relaunch and include HPV vaccination in the national immunization program in Kazakhstan.

## Figures and Tables

**Figure 1 vaccines-10-00824-f001:**
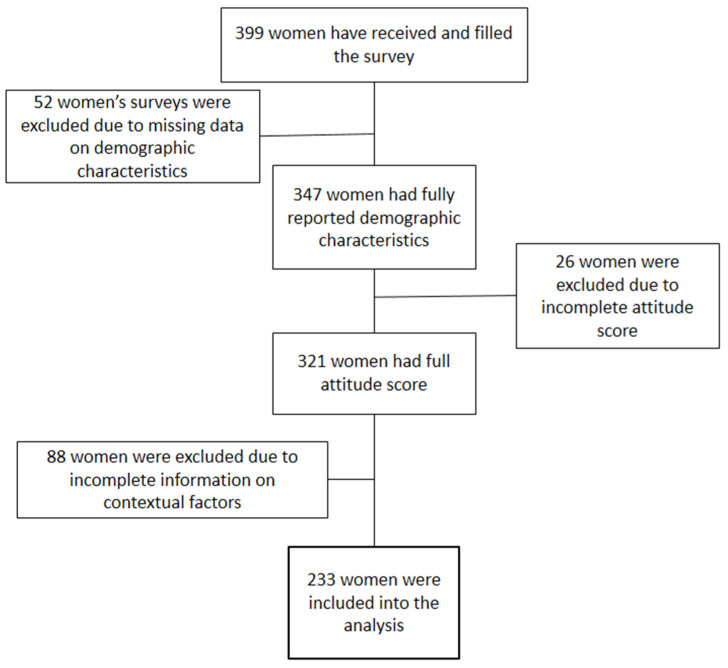
Flow chart on participants’ inclusion.

**Figure 2 vaccines-10-00824-f002:**
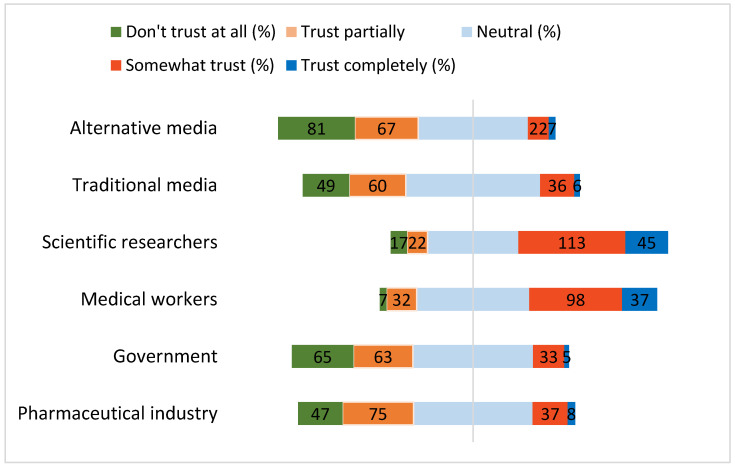
Respondents’ trust to information providers.

**Table 1 vaccines-10-00824-t001:** Social and demographic characteristics of the study participants (N = 347).

Variables	Total N = 347 (%)
Age, Mean 36.46 ± 11.18	
19–27	74 (21%)
28–33	94 (27%)
34–43	86 (25%)
44+	93 (27%)
Ethnicity	
Kazakh	284 (82%)
Other	63 (18%)
Education	
Unfinished/finished school	38 (11%)
College	117 (34%)
University	192 (55%)
City	
Nur-Sultan	86 (25%)
Almaty	80 (23%)
Aktobe	100 (29%)
Oskemen	81 (23%)
Income *	
Low	166 (48%)
Middle	84 (24%)
Upper	97 (28%)
Marital status	
Single	71 (20%)
Not single	276 (80%)
Children	
No children	73 (21%)
1–3 children	239 (69%)
4 or more children	35 (10%)

* Agency for Strategic Planning and Reforms of the Republic of Kazakhstan. Bureau of National Statistics.

**Table 2 vaccines-10-00824-t002:** Contextual factors, N = 233.

Questions	Disagree N (%)	Neutral N (%)	Agree N (%)
Since the controversy over the COVID-19 vaccination, I have less confidence in the HPV vaccination recommendations	88 (38%)	112 (48%)	33 (14%)
Since HPV vaccination program started in Kazakhstan in 2013, had led to complications in few cases, I have less confidence in the health care system	83 (36%)	112 (48%)	38 (16%)
I think vaccinating teenage girls against HPV encourages them to have sex	130 (56%)	80 (34%)	23 (10%)
I am in favor of the compulsory vaccines for children mandated by the government of Kazakhstan	98 (42%)	59 (25%)	76 (33%)
Everyone should be able to decide which vaccines are needed for their children	45 (19%)	103 (44%)	85 (37%)
Alternative medicines strengthen the body’s defenses, thus leading to a complete cure	45 (19%)	47 (20%)	141 (61%)

**Table 3 vaccines-10-00824-t003:** Association between attitude towards HPV vaccination and the trust to the sources, N = 218.

Variables	Attitude towards HPV Vaccination N (%)			*p* Value
	Low 45 (21%)	Middle 63 (29%)	High 110 (50%)	
Pharmaceutical industry				*p* = 0.002 *
Distrust	24 (26%)	15 (16%)	53 (58%)	
Neutral	12 (13%)	39 (43%)	40 (44%)	
Trust	9 (26%)	9 (26%)	17 (48%)	
Government				*p* < 0.001 *
Distrust	25 (26%)	12 (13%)	58 (61%)	
Neutral	12 (12%)	46 (48%)	58 (40%)	
Trust	8 (30%)	5 (18%)	14 (52%)	
Healthcare professional				*p* = 0.004 **
Distrust	8 (36%)	3 (14%)	11 (50%)	
Neutral	17 (19%)	38 (41%)	37 (40%)	
Trust	20 (19%)	22 (21%)	62 (60%)	
Scientific researchers				*p* < 0.001 *
Distrust	4 (15%)	5 (19%)	17 (66%)	
Neutral	17 (24%)	35 (50%)	18 (26%)	
Trust	24 (20%)	23 (19%)	75 (61%)	
Traditional media				*p* < 0.001 *
Distrust	19 (24%)	10 (12%)	52 (64%)	
Neutral	17 (15%)	48 (42%)	48 (43%)	
Trust	9 (37%)	5 (21%)	10 (42%)	
Alternative media				*p* < 0.001 **
Distrust	24 (22%)	17 (15%)	71 (63%)	
Neutral	15 (16%)	44 (48%)	33 (36%)	
Trust	6 (43%)	2 (14%)	6 (42%)	

* *p* value < 0.05, Chi-square test. ** *p* value < 0.05, Fisher’s exact test.

**Table 4 vaccines-10-00824-t004:** Attitude towards HPV vaccination among women including social and demographic characteristics and contextual factors, (N = 233).

Variables	Attitude Prevalence N (%)			*p* Value	Attitude COR (95%)	Attitude AOR (95%) ***
	Negative 43 (18%)	Neutral 70 (30%)	Positive1 20 (52%)			
**Age**						
19–27	7 (14%)	11(23%)	31(63%)	*p* = 0.377	1	1
28–33	16 (24%)	18 (26%)	34 (50%)		0.55 (0.27–1.15)	0.80 (0.36–1.76)
34–43	8 (14%)	20 (35%)	29 (51%)		0.67 (0.32–1.42)	1.23 (0.52–2.93)
44+	12 (20%)	21 (36%)	26 (44%)		0.50 (0.24–1.05)	0.88 (0.37–2.08)
**Ethnicity**						
Kazakh	33 (18%)	60 (32%)	94 (50%)	*p* = 0.381		
Other	10 (22%)	10 (22%)	26 (56%)			
**Education**						
Unfinished/finished school	5 (24%)	13 (62%)	3 (14%)	*p* = 0.001 *	1	1
College	17 (24%)	19 (26%)	36 (50%)		2.29 (0.98–5.34)	1.81 (0.73–4.48)
University	21 (15%)	38 (27%)	81 (58%)		3.40 (1.53–7.56)	2.38 (1.01–5.65)
**City**						
Nur-Sultan	4 (7%)	17 (30%)	35 (63%)	*p* = 0.003 **	1	
Almaty	15 (24%)	13 (21%)	35 (56)		0.60 (0.29–1.23)	
Aktobe	8 (12%)	27 (41%)	31 (47%)		0.57 (0.29–1.14)	
Oskemen	16 (33%)	13 (27%)	19 (40%)			
**Family income**						
Low	26 (27%)	30 (32%)	39 (41%)	*p* = 0.004 **	1	
Middle	5 (7%)	25 (37%)	37 (55%)		2.13 (1.18–3.85)	
Upper	12 (17%)	15(21%)	44 (62%)		2.29 (1.25–4.21)	
**Marital status**						
Single	11 (22%)	12 (25%)	26 (53%)	*p* = 0.548		
Not single	32 (17%)	58 (32%)	94 (51%)			
**Children**						
No children	7 (12%)	9 (16%)	42 (72%)	*p* < 0.001 **	1	1
1–3 children	31 (21%)	45 (30%)	74 (49%)		0.38 (0.20–0.73)	0.45 (0.21–0.94)
4 or more children	5 (20%)	16 (64%)	4 (16%)		0.18 (0.07–0.43)	0.26 (0.10–0.73)
**Effect of COVID-19 vaccination on HPV vaccine perception**						
Disagree	10 (12%)	16 (19%)	57 (69%)	*p* = 0.001 **	1	
Neutral	21 (19%)	43 (38%)	48 (43%)		0.39 (0.22–0.69)	
Agree	12 (32%)	11 (29%)	15 (39%)		0.27 (0.13–0.58)	
**The 2013 HPV vaccination program in Kazakhstan**						
Disagree	9 (10%)	16 (18%)	63 (72%)	*p* < 0.001 **	1	1
Neutral	21 (19%)	45 (40%)	46 (41%)		0.31 (0.18–0.56)	0.34 (0.19–0.62)
Agree	13 (40%)	9 (27%)	11(33%)		0.16 (0.07–0.37)	0.21 (0.09–0.50)
**Relation of HPV vaccination to sexual promiscuity**						
Disagree	21 (16%)	25 (19%)	84 (65%)	*p* < 0.001 **	1	
Neutral	14 (18%)	37 (46%)	29 (36%)		0.42 (0.25–0.71)	
Agree	8 (35%)	8 (35%)	7 (30%)		0.25 (0.22–0.59)	
**Compulsory vaccine mandated by the government**						
Disagree	18 (18%)	27 (28%)	53 (54%)	*p* = 0.037 **	1	
Neutral	9 (15%)	27 (46%)	23 (39%)		0.70 (0.39–1.27)	
Agree	16 (21%)	16 (21%)	44 (58%)		1.08 (0.60–1.94)	
**Freedom of vaccine choice for their children**						
Disagree	4 (9%)	9 (20%)	32 (71%)	*p* < 0.001 **	1	
Neutral	4 (8%)	29 (62%)	14 (30%)		0.29 (0.13–0.66)	
Agree	35 (25%)	32 (23%)	74 (52%)		0.39 (0.19–0.80)	
**Alternative medicine strengthening the body’s defenses**						
Disagree	4 (9%)	8 (18%)	33 (73%)	*p* = 0.015 **	1	1
Neutral	50 (19%)	38 (37%)	45 (44%)		0.31 (0.14–0.65)	0.48 (0.21–1.10)
Agree	19 (22%)	24 (28%)	42 (50%)		0.34 (0.16–0.74)	0.41 (0.18–0.94)

* *p* value < 0.05, Fisher’s exact test. ** *p* value < 0.05, Chi-square test. *** OR adjusted for age, education, number of children, the 2013 HPV vaccination program in Kazakhstan, alternative medicine strengthening the body’s defenses.

## Data Availability

The study questionnaires are available per resonable request form the study PI via email gulzhanat.aimagambetova@nu.edu.kz.

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
