# Peer review of "What Factors Are Associated with Attitudes towards HPV Vaccination among Kazakhstani Women? Exploratory Analysis of Cross-Sectional Survey Data"

_vaccines, 2022, doi:10.3390/vaccines10050824_

Round 1

Reviewer 1 Report

The paper concerns an investigation of women's attitudes towards vaccination against HPV in Kazakhstan. The investigation was made on the basis of a cross-sectional survey among Kazakhstani women and standard statistical methods.

It could be said that it is another paper concerning attidudes towards some vaccination, however, in this case it is the first survey concerning HPV vaccination in Kazakhstan. It is all the more important since there was an unsuccessful HPV vaccination program in Kazakhstan in 2013. So, it is crutial to know what are the current attitudes towards HPV vaccination and what influences these attitudes. It is obvious that such knowledge can be very helpful in preparation of another vaccination program. 

The described in the paper study is an attemp to gather such knowledge and the obtained results are interesting and important.

The Authors wrote about limitations of the performed investigation but there are also some limitations not mentioned by them. The first of these limitations is the fact that the survey was done in four major cities of Kazakhstan. It makes the analyzed sample not representative for the whole Kazakhstani population. The second, not mentioned limitation, is the fact that 55% of the surveyed women have university level education and another 34% of them have college level education. It is another reason that the sample is not representative for the population. So, these limitations should be clearly stated in the paper.

The Authors said that this is a kind of a pilot study, and in my opinion, despite several limitations, it is a valuable research, especially in the light of the fact that it is the first attempt to get knowledge on this topic. 

The Authors mentioned some future continuation of this study. It would be interesting for the readers if they wrote something more about it.

Summarizing, in my opinion the paper is interesting and well wrtten and could be considered for a possible publication in Vaccines after a minor revision.

Author Response

Dear Reviewer,

Thank you very much for the detailed review of our manuscript. We appreciate a lot your time, efforts, and valuable comments and suggestions that helped us to improve the text quality. Please find below our point-by-point responses for all your comments.

Reviewer:

The paper concerns an investigation of women's attitudes towards vaccination against HPV in Kazakhstan. The investigation was made on the basis of a cross-sectional survey among Kazakhstani women and standard statistical methods.

It could be said that it is another paper concerning attidudes towards some vaccination, however, in this case it is the first survey concerning HPV vaccination in Kazakhstan. It is all the more important since there was an unsuccessful HPV vaccination program in Kazakhstan in 2013. So, it is crutial to know what are the current attitudes towards HPV vaccination and what influences these attitudes. It is obvious that such knowledge can be very helpful in preparation of another vaccination program. 

The described in the paper study is an attemp to gather such knowledge and the obtained results are interesting and important.

Response:  Dear Reviewer, thank you very much for your feedback and appreciation of our study.

Reviewer:

The Authors wrote about limitations of the performed investigation but there are also some limitations not mentioned by them. The first of these limitations is the fact that the survey was done in four major cities of Kazakhstan. It makes the analyzed sample not representative for the whole Kazakhstani population.

Response – Dear reviewer, thank you for the comment. We conducted study in four major cities (Nur-Sultan, Almaty, Aktobe, Oskemen), which represent central, southern, western, and eastern regions of Kazakhstan. The corrections detailing it were included in the methods part.  However, we agree that these cities population cannot fully represent the country.

Reviewer:

The second, not mentioned limitation, is the fact that 55% of the surveyed women have university level education and another 34% of them have college level education. It is another reason that the sample is not representative for the population. So, these limitations should be clearly stated in the paper.

Response: Thank you for the comment. We included specific statements in the limitation part.

Study strength and limitations. This was the first study to assess attitude towards HPV vaccination among Kazakhstani women. Another strength of this study is covering female population in the big cities of four Kazakhstani regions (central, southern, eastern, and western) and investigating the factors associated with the attitudes towards HPV vaccination. However, since we approached only the population of big cities, but not these regions’ rural areas, this study cannot fully represent the whole country female population. Moreover, in this study, the proportion of participants with university level and college level education were 55% and 34% respectively. These indicators could be different, if we include participants form rural areas. These are the major limitations, which we hope to overcome in our future studies and to obtain more precise picture in terms of participants’ residence and education. Another limitation of this study is its cross-sectional nature and resulting lack of causality….”

Reviewer:

The Authors said that this is a kind of a pilot study, and in my opinion, despite several limitations, it is a valuable research, especially in the light of the fact that it is the first attempt to get knowledge on this topic. 

Response: Dear Reviewer, Thank you very much for appreciation of our study.

Reviewer:

The Authors mentioned some future continuation of this study. It would be interesting for the readers if they wrote something more about it.

Response: For sure, we will do our best to perform a further investigations on this field  and make it available for the international audience.

Summarizing, in my opinion the paper is interesting and well wrtten and could be considered for a possible publication in Vaccines after a minor revision.

Reviewer 2 Report

Aimagambetova et al’s manuscript describing the factors associated towards HPV vaccination among Kazakhstani women provides insight to their response to a proposed HPV vaccination program and how COVID-19 may have affected attitudes towards vaccination. There are some oversights in the paper that need more detail such as the vaccine introduction attempt in 2013 that was unsuccessful due to complications, but the complications are not described.  Please include more information.  Also, providing more information on the women who receive regular health care is important. Are the women that completed the survey representative of women in Kazakhstan overall?

Methods. 

Is the Questionnaire mentioned in line 90 openly available?  If not, please include as supplemental information. How was it adapted for Kazakhstan?  Can you provide some context on religiosity of the region and how this may affect vaccine uptake? Please provide more information on the “Covid 19 effect” that is mentioned in the manuscript.

Results

  It is mentioned that the majority of women had a university degree.  Is this representative of the nation overall?  If not, please provide a the % of women in Kazakhstan who have a university degree etc….

Table 2 What were the complications in 2013?

Figure 2.  What indicates social media like Facebook/Meta, Instagram etc…?  Is that included in traditional media?  Alternative media? 

Table 3.  This data would be great to compare with another country like the US. 

Table 4 is very busy and should be combined so it fits on 1 page.

Discussion

Why has the rate of cervical cancer been increasing?  Complications and failure are mentioned again, please elaborate.  How would you suggest they  increase the “trustworthy ways” in the vaccine campaign?

Line 254.  What is the MOH?

Line 284.  Are mothers of more children more likely to be less educated?

How can you expand this to study to more women?  The number of women surveyed is minuscule in a country of over 18 million.

Author Response

Dear Reviewer,

Thank you very much for the detailed review of our manuscript. We appreciate a lot your time, efforts, and valuable comments and suggestions that helped us to improve the text quality. Please find below our point-by-point responses for all your comments.

Reviewer

Aimagambetova et al’s manuscript describing the factors associated towards HPV vaccination among Kazakhstani women provides insight to their response to a proposed HPV vaccination program and how COVID-19 may have affected attitudes towards vaccination. There are some oversights in the paper that need more detail such as the vaccine introduction attempt in 2013 that was unsuccessful due to complications, but the complications are not described.  Please include more information.

Response – Thank you for the comment. The details of the complications were provided in the discussion part.

Reviewer

Also, providing more information on the women who receive regular health care is important. Are the women that completed the survey representative of women in Kazakhstan overall?

Response – Dear reviewer, thank you for the comment. We conducted study in four major cities (Nur-Sultan, Almaty, Aktobe, Oskemen), which represent central, southern, western, and eastern regions of Kazakhstan. The corrections detailing it were included in the methods part.  However, we agree that these cities population cannot fully represent the country and will consider it in our future studies.

Reviewer

Methods.

Is the Questionnaire mentioned in line 90 openly available?  If not, please include as supplemental information. How was it adapted for Kazakhstan?  Can you provide some context on religiosity of the region and how this may affect vaccine uptake? Please provide more information on the “Covid 19 effect” that is mentioned in the manuscript.

Response – Thank you for the comment. The questionnaire is available and is attached to the submission as a supplementary file.

According to this source (https://www.gov.kz/memleket/entities/qogam/activities/141?lang=ru) 3834 religious associations are registered in the country, of which 70.3% are Islamic. And according to the 2019 study (https://www.gov.kz/memleket/entities/din/projects/details/729?lang=ru) 92.8% of our population considers themselves religious. We could argue that our population is relatively religious and "strict" religion is dominant in our country. However, since we didn't aim to address whether religion is a factor for HPV vaccine hesitancy I'm not sure how to weave this information into the text.

Reviewer

Results

  It is mentioned that the majority of women had a university degree.  Is this representative of the nation overall?  If not, please provide a the % of women in Kazakhstan who have a university degree etc….

Response – Thank you for the comment.

Most women in the study  (55%) had a university degree, which is in line with the official governmental statistics that report gross enrolment ration in higher education in Kazakhstan in 2020 to be 64.07%, of which 70.5% were women (https://stat.gov.kz/official/industry/62/publication).

Reviewer

Table 2 What were the complications in 2013?

Response – Thank you for the comment. We included details on the complication during  vaccination in 2013 into the  discussion part.

In Kazakhstan, the pushback has intensified after media coverage of two 11-12 years old girls having trouble breathing, hallucinations, and being hospitalized into ICU right after administration of the HPV vaccine. As was later clarified, such reaction resulted from inappropriate actions of medical personnel, who panicked at allergic rash to cold vaccine and who administered a mixture sedatives and pain medications. Nevertheless, the damage was done. Media quoting government officials and medical workers saying, “73 percent of participants in [HPV] vaccine trials acquired new diseases - similar facts have been recorded around the world” and “Who knows what will happen to the reproductive function of [HPV] vaccinated girls in ten years?” (https://www.caravan.kz/gazeta/vakcina-protiv-virusa-papillomy-spasenie-ili-ehksperiment-83376/).”

Reviewer

Figure 2.  What indicates social media like Facebook/Meta, Instagram etc…?  Is that included in traditional media?  Alternative media?

Response – Dear Reviewer, details on the alternative and traditional media sourses are included.

Both alternative (WhatsApp messenger, Instagram, Telegram, YouTube, etc.) and traditional media (TV, newspapers) had lower trust levels among the participants with both negative and positive attitudes towards HPV vaccination.”

Reviewer

Table 3.  This data would be great to compare with another country like the US.

Response – Thank you or the comment. According to the suggestion, this data was compared with US data in the discussion part.

Reviewer

Table 4 is very busy and should be combined so it fits on 1 page.

Response – Dear Reviewer, Thank you very much for your comment. Indeed, table# 4 present a lot of information, which we would like to keep for a potential readers.  We hope for your understanding.

Reviewer

Discussion

Why has the rate of cervical cancer been increasing? 

Response – Dear Reviewer, thank you for the comment and your interest to the situation. There is no direct answer to the question. The increasing rates of cervical cancer is reported in Kazakhstan, but the healthcare authority do not provide reports on the factors and reasons leading to it. As a team, working for many years on the topic and analyzing many data, we assume that the rates of the cervical cancer is increasing in Kazakhstan due to the infective screening program with a low coverage. We included this information in the introduction part.

Reviewer

Complications and failure are mentioned again, please elaborate.  How would you suggest they  increase the “trustworthy ways” in the vaccine campaign?

Response – As we highlighted in the discussion part, “the fact that in our study researchers and healthcare professionals are the most trusted source of information about the vaccine” suggests that  Ministry of Healthcare can work via the healthcare professionals to increase the HPV vaccines awareness and improve the attitudes.

Reviewer

Line 254.  What is the MOH?

Response – MOH is and abbreviation – the Ministry of Healthcare, was replaced by the full words.

Reviewer

Line 284.  Are mothers of more children more likely to be less educated?

Response – Dear reviewer, thank you for the comment. Chi-squared test showed correlation between the number of children and level of education of the mother (p-value=0.043), where only 36% mothers with 4 or more children obtained higher education. However, the interaction between number of children and education was not significant and the model was better without the interaction term. Therefore, we did not include it into the manuscript.

Reviewer

How can you expand this to study to more women?  The number of women surveyed is minuscule in a country of over 18 million.

Response – Dear Reviewer, Thank you for the comment. This was a pilot study. The proportion of women in reproductive age is around 25%. This will be a task for future studies that we plan to create and apply for a bigger project to cover as many women as possible.

Reviewer 3 Report

Thank you for giving me the opportunity to review the manuscript by G. Aimagambetova et al. ‘What factors are associated with attitudes towards HPV vaccination among Kazakhstani women? Exploratory analysis of cross-sectional survey data’

It was my pleasure to read your article and the work done is appreciated.

This study shows contrary attitudes towards HPV vaccination among Kazakhstani women. It is notable that almost 50% of women having negative or neutral attitude towards the anti-HPV vaccine. The need for introduction of HPV educational interventions to relaunch and include HPV vaccination in the national immunization program in Kazakhstan is mandatory.

The introduction is clear in its structure and in the references of the studies consulted.

Material and methods as well as results sections are clear.

Discussion section is written well however I have some suggestions to the authors.

Please write a paragraph in the discussion section including additional positive effects of HPV vaccination in population with cervical pathology and how knowledge and behaviour regarding gynaecological pathologies can affect HPV status.

Please also add the following references:

  1. Valasoulis, G.; Pouliakis, A.; Michail, G.; Kottaridi, C.; Spathis, A.; Kyrgiou, M.; Paraskevaidis, E.; Daponte, A. Alterations of HPV-Related Biomarkers after Prophylactic HPV Vaccination. A Prospective Pilot Observational Study in Greek Women. Cancers 2020, 12, 1164. https://doi.org/10.3390/cancers12051164
  2. Tsakiroglou M, Bakalis M, Valasoulis G, Paschopoulos M, Koliopoulos G, Paraskevaidis E. Women's knowledge and utilization of gynecological cancer prevention services in the Northwest of Greece. Eur J Gynaecol Oncol. 2011;32(2):178-81. PMID: 21614908
  3. Valasoulis, G.; Pouliakis, A.; Michail, G.; Daponte, A.-I.; Galazios, G.; Panayiotides, I.G.; Daponte, A. The Influence of Sexual Behavior and Demographic Characteristics in the Expression of HPV-Related Biomarkers in a Colposcopy Population of Reproductive Age Greek Women. Biology 2021, 10, 713. https://doi.org/10.3390/biology10080713

Author Response

Dear Reviewer,

Thank you very much for the detailed review of our manuscript. We appreciate a lot your time, efforts, and valuable comments and suggestions that helped us to improve the text quality. Please find below our point-by-point responses for all your comments.

Reviewer:

Thank you for giving me the opportunity to review the manuscript by G. Aimagambetova et al. ‘What factors are associated with attitudes towards HPV vaccination among Kazakhstani women? Exploratory analysis of cross-sectional survey data’

It was my pleasure to read your article and the work done is appreciated.

This study shows contrary attitudes towards HPV vaccination among Kazakhstani women. It is notable that almost 50% of women having negative or neutral attitude towards the anti-HPV vaccine. The need for introduction of HPV educational interventions to relaunch and include HPV vaccination in the national immunization program in Kazakhstan is mandatory.

Response:  Dear Reviewer, thank you very much for your feedback and appreciation of our study.

Reviewer:

 The introduction is clear in its structure and in the references of the studies consulted.

Material and methods as well as results sections are clear.

Response:  Dear Reviewer, thank you very much for your feedback.

Reviewer:

Discussion section is written well however I have some suggestions to the authors.

Please write a paragraph in the discussion section including additional positive effects of HPV vaccination in population with cervical pathology and how knowledge and behaviour regarding gynaecological pathologies can affect HPV status.

Please also add the following references:

  1. Valasoulis, G.; Pouliakis, A.; Michail, G.; Kottaridi, C.; Spathis, A.; Kyrgiou, M.; Paraskevaidis, E.; Daponte, A. Alterations of HPV-Related Biomarkers after Prophylactic HPV Vaccination. A Prospective Pilot Observational Study in Greek Women. Cancers 2020, 12, 1164. https://doi.org/10.3390/cancers12051164
  2. Tsakiroglou M, Bakalis M, Valasoulis G, Paschopoulos M, Koliopoulos G, Paraskevaidis E. Women's knowledge and utilization of gynecological cancer prevention services in the Northwest of Greece. Eur J Gynaecol Oncol. 2011;32(2):178-81. PMID: 21614908
  3. Valasoulis, G.; Pouliakis, A.; Michail, G.; Daponte, A.-I.; Galazios, G.; Panayiotides, I.G.; Daponte, A. The Influence of Sexual Behavior and Demographic Characteristics in the Expression of HPV-Related Biomarkers in a Colposcopy Population of Reproductive Age Greek Women. Biology 2021, 10, 713. https://doi.org/10.3390/biology10080713

Response:  Dear Reviewer, thank you very much for your comment.

A paragraph addressing the mentioned information and citing the suggested papers were included in the text of the discussion.

Reviewer 4 Report

Since the 1970’s, evidence has been emerging causally linking HPV with different human neoplastic lesions. Until now, 200 HPV types have been fully characterized, comprising both cutaneous HPV types causing benign clinical manifestations known as skin warts (papilloma), and mucosal HPV types inducing benign papilloma. Cervical cancer (CC)  causes significant morbidity and mortality worldwide.

The HPV infection is associated with the invasive cancer in the anogenital mucosa as well as in the respiratory (sinonasal, larynx, trachea, bronchus) and upper digestive tract (oral mucosa, oropharynx, esophagus. The currently available vaccines against two HPV types (HPV 16 and HPV18) have the potential of reducing the incidence of cervical and anogenital cancer. HPV vaccines have also been found to reduce infections in other tissues that HPV infects.

Therefore, knowledge about vaccination in the population is very important. The Authors wanted to know the causes of non-vaccination in the Kazakh population.The studies are interesting, well statistically analyzed. However, it seems to me that it would be better to present the problem in a broader context. The idea is to point out that a persistent HPV infection can lead to the development of other cancers as well.  My minor comments are following:Ad. Introduction1.The Authors focused only on the role of HPV in cervical cancer. The relationship of HPV infection with other cancers is also well documented. For this reason, it would be good to supplement the information that HPV infection plays a role in other cancers, e.g. anogenital tumours, head and neck cancers (Jalouli, Syrjanen, Polz-Dacewicz)Therefore, immunization can protect against the development of other cancers as well.2.The Authors emphasize the high frequency of HPV infections in Kazakhstan. Therefore, it would be good to compare this with the situation in other countries. Ad. ResultsDid the Authors also analyze whether the respondents are  against other vaccinations as well ?  

Author Response

Dear Reviewer,

Thank you very much for the detailed review of our manuscript. We appreciate a lot your time, efforts, and valuable comments and suggestions that helped us to improve the text quality. Please find below our point-by-point responses for all your comments.

Reviewer

Since the 1970’s, evidence has been emerging causally linking HPV with different human neoplastic lesions. Until now, 200 HPV types have been fully characterized, comprising both cutaneous HPV types causing benign clinical manifestations known as skin warts (papilloma), and mucosal HPV types inducing benign papilloma. Cervical cancer (CC)  causes significant morbidity and mortality worldwide.

The HPV infection is associated with the invasive cancer in the anogenital mucosa as well as in the respiratory (sinonasal, larynx, trachea, bronchus) and upper digestive tract (oral mucosa, oropharynx, esophagus. The currently available vaccines against two HPV types (HPV 16 and HPV18) have the potential of reducing the incidence of cervical and anogenital cancer. HPV vaccines have also been found to reduce infections in other tissues that HPV infects.

Therefore, knowledge about vaccination in the population is very important. The Authors wanted to know the causes of non-vaccination in the Kazakh population.The studies are interesting, well statistically analyzed. However, it seems to me that it would be better to present the problem in a broader context. The idea is to point out that a persistent HPV infection can lead to the development of other cancers as well.  

My minor comments are following:

Ad. Introduction1.The Authors focused only on the role of HPV in cervical cancer. The relationship of HPV infection with other cancers is also well documented. For this reason, it would be good to supplement the information that HPV infection plays a role in other cancers, e.g. anogenital tumours, head and neck cancers (Jalouli, Syrjanen, Polz-Dacewicz)Therefore, immunization can protect against the development of other cancers as well.

Response: Dear Reviewer, thank you or the comment. The suggested information was included into the introduction part and the suggested sources were cited in the reference list.

Reviewer

2.The Authors emphasize the high frequency of HPV infections in Kazakhstan. Therefore, it would be good to compare this with the situation in other countries. 

Response – The worldwide rates of HPV were provided in the introduction part.

Reviewer –

Ad. Results Did the Authors also analyze whether the respondents are  against other vaccinations as well ?  

Response  - Thank you for the comment. In this study we did not aim to investigate attitudes towards other vaccines in Kazakhstan. The study on this tipoc was done by our colleagues and already published.

Issanov A, Akhmetzhanova Z, Riethmacher D, Aljofan M. Knowledge, attitude, and practice toward COVID-19 vaccination in Kazakhstan: a cross-sectional study. Hum Vaccin Immunother. 2021;17(10):3394-3400. doi:10.1080/21645515.2021.1925054
